March 22, 2024

# Retraction Notice

Retraction: The Publisher was alerted that prior to the submission of this article, the identical article had been submitted to other journals by authors other than Ke Zhang. We confirmed this was the case and we reached out to Ke Zhang for an explanation. We have received no reply from Ke Zhang and so we are forced to assume that this article was inappropriately copied by someone who had access to it via one of the earlier submissions. This publication is therefore Retracted and we have reported this situation to JiuZhou Polytechnic.

PeerJ Editorial Office. 2024. Retraction: A wormhole attack detection method for tactical wireless sensor networks. PeerJ Computer Science 10:e1449/retraction https://doi.org/10.7717/peerj-cs.1449/retraction



# A wormhole attack detection method for tactical wireless sensor networks

Ke Zhang

Academic Affairs Office, JiuZhou Polytechnic, Xuzhou, China

## ABSTRACT

Wireless sensor networks (WSNs) are networks formed by organizing and combining tens of thousands of sensor nodes freely through wireless communication technology. WSNs are commonly affected by various attacks, such as identity theft, black holes, wormholes, protocol spoofing, *etc*. As one of the more severe threats, wormholes create passive attacks that are hard to detect and eliminate. Since WSN is often used in the tactical network field, a planned secure network is essential for military applications with high security. Guard nodes are traffic monitoring nodes used to supervise neighbors' data communication around the tactical networks. Therefore, this work proposes a Quality of Service (QoS) security mechanism to select multiple dual-layer guard nodes at different paths of the WSN based on the path qualities to detect wormholes. The entire network's links are categorized into high, normal, and low priority levels. As such, this study aimed to confirm the security of high priority nodes and links in the tactical network, avoid excessive overhead, and provide random security facilities to all nodes. The proposed measures of the QoS-based security provision, including link cluster formation, guard node selection, authenticated guard node identification, and intrusion detection, ensure economic and efficient network communication with different quality levels.

## INTRODUCTION

Wireless security networks (WSNs) are a type of sensor network that deploys multiple wireless sensor nodes around vast geographical regions. Each sensor node in the network can be manufactured with multiple sensors and transceivers that observe environmental objects, such as heat, moisture, pressure, *etc*. These sensor nodes and other gateway systems are implemented into tactical networks to increase security. However, despite the secure environment, the network is still vulnerable to different attacks that harm its functions and, thus, requires new techniques to enhance security further. Guard nodes provide a security technique that employs separate nodes with other tactical nodes to supervise neighbours' activities. These nodes must be optimally selected from any part of the tactical WSN. to maintain data transmission quality. In this work, guard nodes were chosen for detecting wormhole attacks raised in the tactical WSN.

Corresponding author
Ke Zhang, zhangke0778@163.com

A wormhole attack is a passive attack that creates a separate unauthorized link between two communicating nodes, subsequently causing the legitimate nodes to send data through this link. Thus, the data are intercepted, and the corresponding nodes are compromised. In a tactical WSN, the guard node selection should be based on current link qualities and priorities to ensure dedicated security benefits. However, in data communication, QoS in security depends on its availability, reliability, and serviceability. This work analyzed the specific characteristics, limitations, and QoS factors of tactical WSNs. Specifically, guard nodes were selected for monitoring purposes by analysing the quality metrics of each node.

Because selecting multiple guard nodes helps to protect the links and nodes from wormhole attacks. The principle of QoS is to ensure the security of all links and guard node activities. At the same time, the priority-based security provision is achieved using this effective guard node selection process. To develop QoS-based guard nodes and establish secure monitoring to eliminate wormholes, supervised monitoring nodes (SMNs) are improved with quality control techniques. *Ji, Chen & Zhong (2015)* analysed wormhole attackers and the complexity of intrusion detection, focusing on QoS and security difficulties. Results showed that such attacks do not concentrate on network availability, communication reliability, or security in WSN. Therefore, the prior wormhole attack detection and performance estimation was the target of the research.

In this work, guard node-based security was combined with QoS for a WSN as a protective measure against wormhole attacks due to inadequate resources and composite interactions of WSN operations. This proposed measure is pertinent for developing secured and QoS-provisioned data transmission.

This presented work suggests the selection of multiple guard nodes to protect the tactical WSN from wormhole attacks. Each node in the network is equipped with wormhole detection procedures and alarm units and can be requested to act as a guard node. At the same time, the guard node uses wormhole detection procedures to detect attacks from the nearest hops. In addition, the proposed system ensures link quality, priority, and QoS levels of each node to provide guard node-based security. The rest of this article is organized as follows. The second section presents related works. The third section describes QoS-based guard node selection procedures, followed by discussions on QoS attainments, secure guard node authentication, and wormhole detection procedures. The fourth section explains the results of implementing the proposed QWDGN. The fifth section provides concluding remarks.

## RELATED WORK

Various detection techniques have been proposed to protect tactical WSNs against attacks. Precisely, the wormhole attack detection techniques follow either rule-based, feature-based, or signature-based identification. *Karthigadevi, Balamurali & Venkatesulu (2018a)* proposed a transmission round time-oriented wormhole attack detection algorithm in wired networks. In this approach, the authors analysed traffic attributes using an enhanced interior gateway routing protocol then implemented the proposed mechanism in cisco-based networks. *Lakshmi & Yadav (2019)* discussed effective wormhole detection

strategies in wireless mesh networks containing more links than other networks. The authors concentrated on mesh network links in which wormholes were created to misuse the network resources. The authors further investigated the performance rates of attack detection algorithms against a large number of wormhole links.

*Ji, Chen & Zhong (2015)* established a distributed-level coding system, DAWN, to protect the network resources against multiple wormhole attacker nodes. The proposed system was designed to monitor local, and global network services, communication synchronization patterns, and wormhole attacks by controlling network coding protocols. Both *Khan & Lavagno (2012)* and *Perillo & Heinzelman (2005)* analysed proactive and reactive protocols for WSN security, focusing on support for data communication, security problems, and the necessary solutions.

*Tun & Maw (2008)* and *Kurmi, Verma & Soni (2017a)* proposed wormhole attack detection strategies in WSNs to protect the nodes. In these works, wormhole attacks were first created by multiple attackers among legitimate nodes, then detected using the proposed algorithms based on the node signatures and transmission patterns. However, the proposed strategies suffer from a limited number of nodes. In the same way, *Kumhar & Ukani (2015)* proposed QoS support protocols for WSNs to build reliable security in the proposed system. Moreover, *Wang et al. (2010)* devised a QoS-based efficient routing approach using MAC optimization for cross-layer activities between MAC and routing protocol procedures. The authors claim that this design performs better in WSN, whereby the MAC and IP data sequences can be customized to obey QoS requirements. Further, the authors suggest that the QoS-based routing strategy offers less delay and smaller packet loss compared to other methods.

Previous studies have proposed diverse viewpoints and principles of QoS. For instance, *Dutta & Dunkels (2012)* investigated QoS measurements and optimal power consumption using QoS procedures and policies with the Gur prototype, and found that node dimension, deployment, coverage, and liveliness of each node constrain QoS. Dissimilar to other works, this work developed QoS as a provision to guard node selection procedures. The researchers matched the requirements of both QoS and security in a single view to building an efficient guard node-based Intrusion Detection System (IDS). Since tactical WSNs are prone to different types of wormhole attacks, *Roy & Khan (2020)* aimed to establish security against wormhole attacks without compromising the quality performance of the network. The latter authors incorporated security policies and a QoS strategy as a protective measure against wormhole attacks.

Similarly, *Pazynyuk et al. (2008)* investigated the challenges and needs of QoS-based security features by selecting guard nodes and using SMNs to maintain the optimal data loss rate, transmission rate, and delay with energy utilization. Because QoS and security concerns are not associated with the context of WSN, *Karthigadevi, Balamurali & Venkatesulu (2018a)* proposed different solutions against wormhole attacks. The authors emphasized the WSN environment, where both QoS and security attainments are required for successful operation. Tactical networks, such as those used in military target tracking applications, are expected to be more secured. *Kurmi, Verma & Soni (2017)* demonstrated the impact of the inclusion of certain security protocols on the rate of QoS and the effects

of specific QoS parameters on the network's security needs. All mentioned studies found notable correlations between QoS and security despite not being closely related notions.

Even though there are different routing and wormhole attack detection techniques, security and QoS maintenance are vital in WSNs (*Poonam & Preeti, 2014*). In *Fathi et al. (2021)* the quality of service-based sensor location modelling is done, and the service pricing is determined based on that *Sartori & Melen (2021)* describes the usage of wireless sensor networks with mobile elements for designing a platform for the smart environment. In this manner, this presented work suggests security provisions against wormhole attacks (*Kaliyar et al., 2020*; *Karthigadevi, Balamurali & Venkatesulu, 2018a*) by considering link-wise QoS requirements and offering the flexibility of customized security metric allotments for different links in tactical WSNs. Specifically, QoS' needs and security qualities are interrelated and secured, while security levels are ensured using Hash-based Authentication Codes (HMAC) computations. Herein, the communication parameters were optimized to enhance the security of QoS levels.

Based on the analysis, both the computation and communication overhead provide efficient security procedures and, thus, improved security for nominal computation overhead in the WSN.

# QOS-BASED WORMHOLE DETECTION USING GUARD NODES

In the proposed QWDGN, security and QoS parameters were analyzed and correlated. First, security levels were determined with varying HMAC complexities and packet lengths, as given in Table 1. Then, QoS and guard node characteristics were mapped for the selection procedures. In the next phase, the selected guard nodes were identified. In the end, the wormhole attack detection procedures were activated to ensure security in the tactical WSN.

Providing security in the selection process of guard nodes creates additional control messages and data exchanges. This can cause the packet length to increase, affecting the QoS assigned to the data transmission.

Table 1 provides details of the various security levels. Discussion of related works focuses on the secured packets with the help of HMAC. As the size of HMAC increases, the packet length also increases, which affects data transmission. Thus, building guard node-based IDS with minimal cost of packet overhead is an important task. Only a few research works describe object or node-tracking mechanisms in WSNs.

## Guard nodes and master nodes

Guard nodes contain the intrusion detection system (IDS) as the active agent to configure rules and procedures to validate node traffic. Since the IDS agent can be designed to monitor any violations in the tactical network, it was configured with wormhole attack patterns in this work. Guard nodes in the network can activate these IDS procedures to validate their behaviors and traffic patterns. Figure 1 shows a sample tactical WSN with various clusters in which several nodes can be grouped. Guard nodes must be selected

**Table 1  Security levels and packet lengths.**

| Security levels | Security details | Packet length (Bytes) |
| --- | --- | --- |
| 0 | Not secured packet | 54 |
| 1 | 8 bytes of HMAC | 62 |
| 2 | 12 bytes of HMAC | 66 |
| 3 | 16 bytes of HMAC | 70 |
| 4 | 20 bytes of HMAC | 74 |

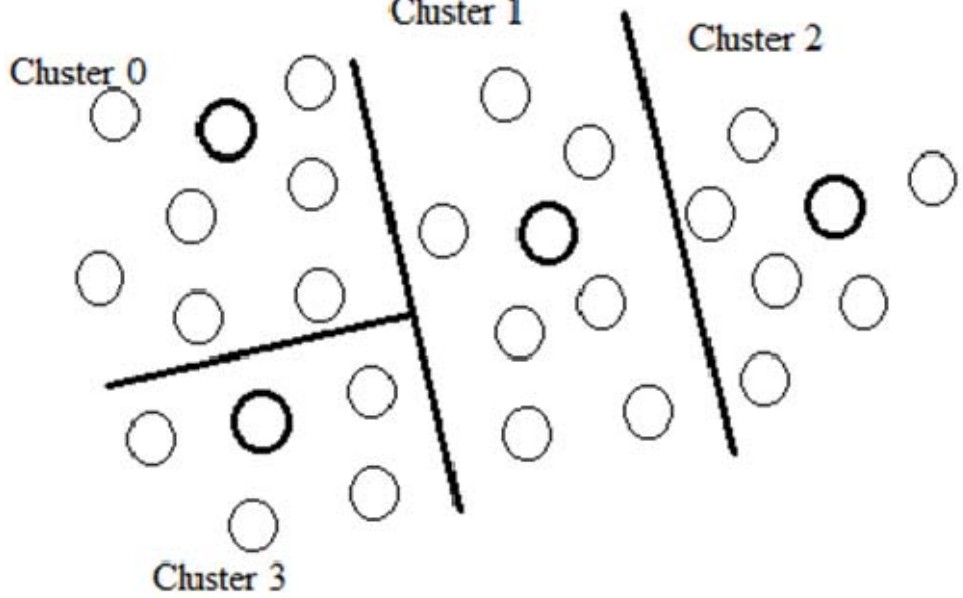

**Figure 1  Tactical WSN formation.**

based on link efficiency to keep QoS constraint. In addition, the supervising master nodes (SMNs) were configured to organize guard nodes' activities and reports.

Figure 1 also illustrates the guard node selection process in the initial steps of the selection algorithm. In the network setup stage, secure cluster formation and guard node selection phases were incorporated with QoS constraints. The following section explains QoS-based guard node selection algorithms and QoS-based SMN configuration procedures.

As seen in Fig. 1, the network setup displays four clusters, each containing members, cluster heads, and SMNs. Algorithm 1 represents the QoS-based cluster formation and cluster head election, assuming there are four classes of QoS requirements.

## QoS-based cluster formation

QoS attributes, such as delay, bandwidth, jitter, energy, and throughput, are given reference values to classify clusters and node characteristics. Algorithm 1 (as shown in Fig. 2) requires a few additional steps for incorporating QoS requirements in each cluster. In this setup phase, quality measurements for all nodes and links are defined and updated in the base

**ALGORITIIM 1: Cluster Formation**

Required: Data set

Input: nodes in network, Q - QoS data requirements

Output: 'n' clusters

-------------------------------------+-------------------------------------

Begin

Step 1: Initialize network nodes and locations

Step 2: Determine Euclidean distance between nodes

Step 3: Generate clusters of sensor nodes

Step 4: Execute cluster head identification routine

Step 5: Evaluate optimal distance threshold values

Step 6: Construct Minimum Spanning Tree

Step 7: Read Q attributes (given in Table 2)

Step 8: Update the link priorities and node priorities to cluster   heads

Step 9: Cluster heads updates priority vector into base station

Step 10: Redo the setup phase periodically.

**Figure 2** The pseudo-code of Algorithm 1.

**Table 2** Simulation attributes.

| Simulator name | NS-3 |
|---|---|
| Simulation Time | 200 s |
| Area | 1000*1000 m |
| Number of nodes | 300 |
| Power (Joules) | 200 |
| Transmitting power (Joules) | 0.5 |
| Routing protocol | AOMDV |

station and the cluster head. The quality definitions are provided in Table 2, in which the dots specify the range or sequence of link quality.

Figure 3 shows the clusters and cluster node links. Using these nodes and QoS requirements, the guard nodes must be selected, while SMNs need to be configured to monitor the nodes. Figure 4 displays the creation of wormhole nodes and links (grey

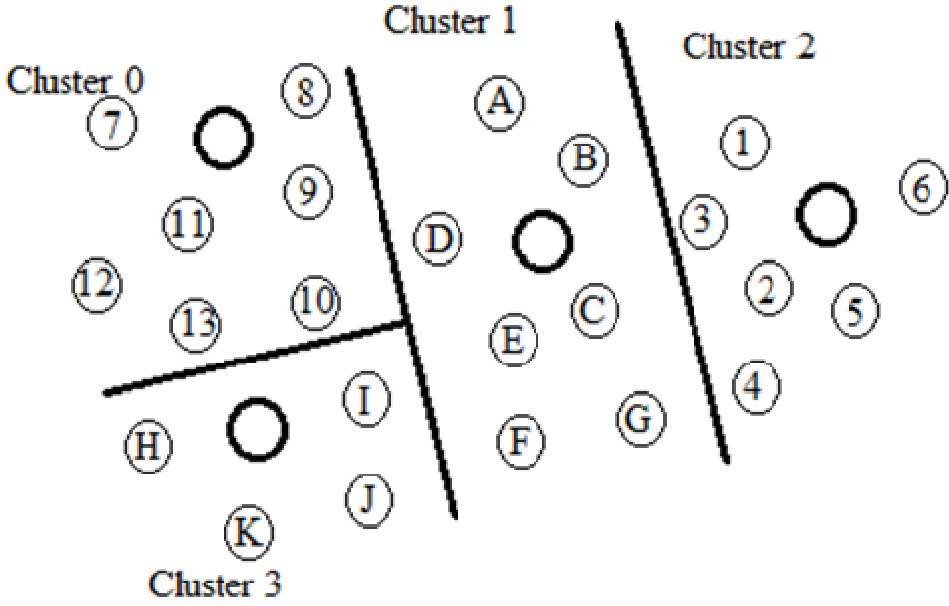

**Figure 3** Clusters, nodes, and link identification.

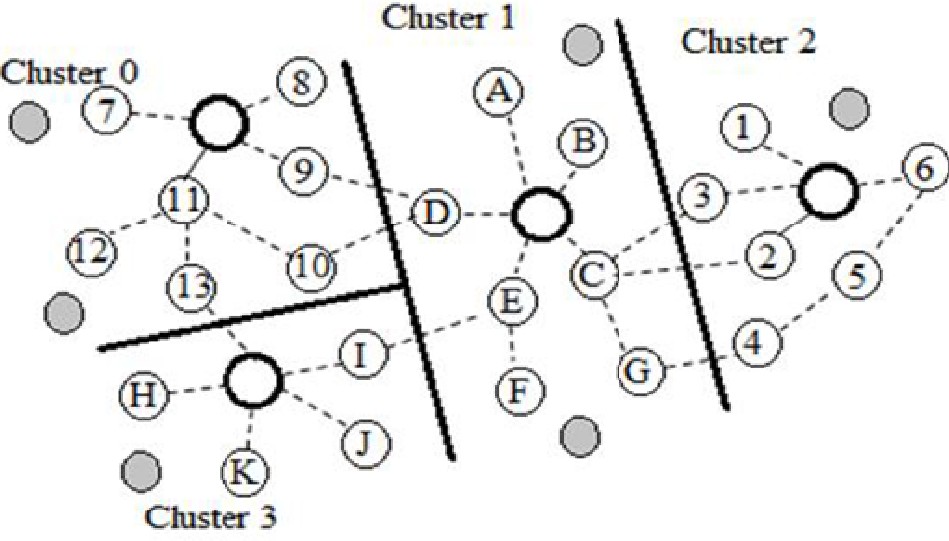

**Figure 4** Different types of wormholes in each cluster.

nodes) in different clusters, in which the short wormhole are affected by Node 1 and Node 6 from Cluster 2.

Besides Nodes 1 and 6 from Cluster 2, Nodes H and K from Cluster 3 are also affected by short wormhole attackers. A single wormhole attacker can use high transmission energy to entice the neighbour nodes to send their data to the attacker. This creates a wormhole

between Nodes 1 and 2 and between Nodes L and M, presenting the first case of vulnerability creation. In comparison, Nodes 7, 15, 14, 13, and 12 from Clusters 0 and 1 are influenced by the longest wormholes, which affect the entire link.

According to the QoS requirements and security needs, the proposed QWDGN provides controllable security complexities. For example, level 0 requires high priority of QoS but allows minimal security overhead, thus, QoS and security levels are indirectly proportional. Yet, this lack of security is not allowable in the best QoS links. In Cluster 1, the affected nodes include Nodes B, C, G, and the cluster head. All wormhole attackers collect packets from their victims and then misuse or divert the collected information. Otherwise, they may alter the received data and send false information to other network nodes. These attacks create extensive issues for low, medium, and high-priority links.

Due to the complex nature of WSNs, identifying guard nodes and maintaining the availability of nodes are complicated tasks. Therefore, in addition to the optimal guard node selection algorithms, this presented study performed the secure cluster head identification and trust evaluation procedures, as reported in previous works.

## QoS-based guard node selection

Algorithm 2 (as shown in Fig. 5) is responsible for validating each link in the network and then subsequently analyzing the priority vector details, which describe the priority levels of each link. From the initial level of analysis, the possible guard nodes are identified. Once the list of optimal neighbours has been formed, they can share their monitor requests with the nodes to be monitored. Advanced Encryption Standard (AES) encrypts these requests at each guard node. This is explained in Algorithm 3 (as shown in Fig. 6).

## Authenticating guard node

The nodes that receive valid permission from Node 'i' can monitor the guard node. Typically, the guard nodes are not selected from high priority QoS links, but rather from low priority links, which causes delays in sensitive data transmission. Also, the bandwidth of superior priority links is saved.

By using this approach, throughput is increased. For MP links, the guard nodes are selected based on the W weight vector, which may fractionally vary between 0 and 1. Once these guard nodes are selected successfully, they are under the control of SMNs. In this QoS-based security mechanism, the SMNs need to be configured in different manners.

## SMN configuration

SMN is the master node that acts as the authenticated cluster head and must be configured for maintaining the operation of guard nodes in wormhole detection. Guard nodes detect wormholes at their nearest hops, then raise alarms and send reports to their closest SMNs, which repeat the wormhole alert messages to all other members of the WSN cluster. Algorithm 4 (as shown in Fig. 7) gives the details of SMN configuration. Figure 8 illustrates the creation and utilization of guard nodes and SMNs, represented by shaded and bold line circles, respectively. The lines between circles indicate the monitoring processes between the grey nodes in the four clusters with several attackers (circles). In Cluster 0, Node 8

**Algorithm 2: Guard Node Selection**

Input: 'c' Cluster nodes, Q attributes
Output: 'o' optimal neighbours

-------------------------------------------------------------------

Step 1: Identify 'c' cluster nodes
Step 2: Compute signature pairs (sa, sb)
Step 3: Validate the signatures with their neighbours
Step 4: Form neighbour list, NList, at each node
(valid neighbours)
Step 5: Compute D-> I (NList), ER-> I (NList),
NN-> I (NList), and ND-> I (NList)
Where i = ith monitored node in the network
  D = Distance (between the nodes)
  ER = Remaining energy of neighbours of ith node
  NN = Neighbour of neighbours of ith node
  ND = Node degree (number of connections
    for a node) of neighbours of ith node
  NList = Neighbour list of ith node in the network
Step 6: Compute the ranges of ER, NN and ND
Step 7: Identify the QoS priority levels,
P_Qlink -> HP||MP||LP
Step 8: Set HP ->H_Security, MP->M_Security and
LP->L_Security
Step 9: Find 'o' optimal neighbours and eliminate others
Step 10: Allot H_Security : $1 \leq o \leq 3$;
M_Security : $1 \leq o \leq 2$; L_Security : $1 \leq o$
Step 11: Set ID sensitive promiscuous mode count to 'o'

-------------------------------------------------------------------

**Figure 5** **The pseudo-code of Algorithm 2.**

monitors Node 9, while Node 10 monitors Node 13. Assuming that Node 6 is sending data to any other node in a cluster, then that data transmission is monitored by two nodes. The monitoring processes are dependent on QoS conditions. For example, if Node 3 withdraws its monitoring task on Node 6 to monitor Node 1, it is assumed that the association between Nodes 1 and 3 has higher priority than between Nodes 3 and 6. The same situation can occur for the operation of Node 5 and each cluster to maintain

**Algorithm 3: Authenticating Guard Nodes**

Input: 'o' optimal neighbours

Output: 'Cp' cluster guard nodes

-------------------------------------------------------------------

Step 1: Identify the list of 'o' neighbours, LNo

Step 2: Find neighbours G1->LNo∉L_Qlink; Update G1 in all cl

where L_Qlink = participants of Qlink

Step 3: Compute request for monitoring process,

mreq=SAESK(hop_distance‖TTL‖RTT‖Timestamp‖

Requesti ng key‖P_Qlink‖HMAC)

Step 4: G1 sends monitoring request (mreq) to ith

Node (monitored node) if and only if

   G1->LNo∉L_Qlink / LNo in Qlink but idle

Step 5: Set G1->LNo∉L_Qlink (HP), G1->LNo may

be in Qlink (LP), LNo∉L_Qlink*W (MP);

   W = traffic weight factor

Step 6: Node 'i' decrypts mreq using shared SAES key, K

Step 7: Secured mreq has been evaluated at ith node

Step 8: Node 'i' computes monitor response,

mres=SAESK (Vbit‖PK ‖ Gt ‖ hop_distance ‖

Timestamp ‖ nonce ‖ HMAC)

   where TTL = Time to live

   RTT = Round-trip time

 Vbit = Validity bit (1 - for valid guard nodes;

  0 - for invalid nodes)

   PK = Permission key for guard nodes

   Gt = Guard node timer

Step 9: Node 'i' sends mres to request nodes, G1

Step 10: Cp guard nodes receive valid reply from ith node

-------------------------------------------------------------------

**Figure 6** **The pseudo-code of Algorithm 3.**

the network traffic requirements based on QoS metrics. Once this QoS-based intrusion detection is enabled in the respective guard node, the node executes its IDS agent module.

## Wormhole attack detection using guard nodes

Usually, wormholes create separate attacker nodes at the transmitting ends (source and destination). The wormhole attackers generate a private unauthenticated link from the source to the destination to intercept the data. Since this attack is not openly active, it

**Algorithm 4: SMN Configuration**

Input: 'n' cluster heads

Output: Supervising master node activation

-------------------------------------------------------------------

Step 1: Identify 'n' cluster heads
Step 2: Set dual mode operation flag, DF = 1 (cluster head acts as SMN)
Step 3: Set time slots for multiple guard nodes, if both guard nodes monitor same neighbour node
Step 4: Find P_Qlink of cluster communication
Step 5: Execute priority scheduling-based monitoring process
Step 6: G is allowed to monitor neighbours in order:
    HP neighbours > MP neighbours > LP neighbours
Step 7:    G_ LP link is allowed to monitor any neighbour
    G_MP link is allowed to monitor the neighbour w.r.t W
    G_HP link is identified if HP link is idle
Step 8:    G monitors LP link nodes if and only if HP & MP are idle
Step 9:    SMN receives M.Report (monitor report) from guard nodes and analyses the report
Step 10: SMN monitors the traffic of G set of a cluster
Step 11: SMN updates the wormhole alert details, A.Report (attack report), to other clusters and base station
Step 12: Set DF = 0 (act as usual cluster head)
Step 13: Repeat the process

-------------------------------------------------------------------

**Figure 7** The pseudo-code of the Algorithm 4.

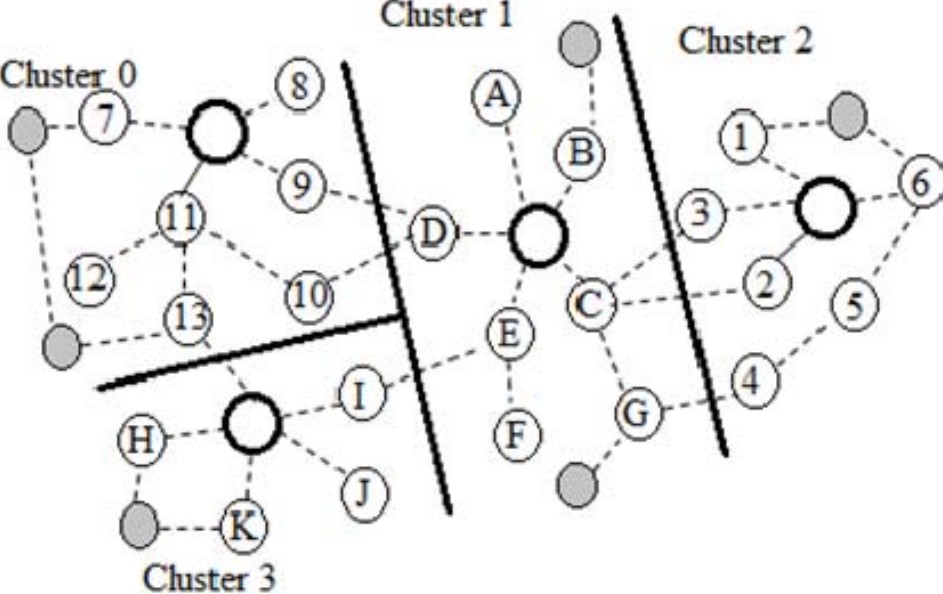

**Figure 8** Guard nodes (shaded circles) and SMNs (bold line circles) in cluster neighbors.

**Algorithm 5: Attack Detection**

Input: Transmitted packets, QoS attributes
Output: Detection of wormhole attacks
-----------------------------------------------------------------

Step 1: Start the timer, GT; G starts snooping neighbour node's transmission
Step 2: Set QoS priority timer 'PT' (monitors until 'PT' expires)

Step 3: Interpret the packets using WPK

Step 4: Checks classified traffic attributes for ((DoS_WH1) && (DoS_WH2) && (DoS_WH3) &&
           (DoS_WH4)= =Match_Null)
Step 5: Checks if ((Sigi&& LC== Valid),
           where LCi = Location coordinates of monitored node
Step 6: Check (Attack_type==Normal)
Step 7: Confirm that the node and data are genuine? and the   data are genuine
Step 8: Guard node resets timers, GT and PT
Step 9: Generate A.Report and transmit the report to cluster members as well as to the base station
Step 10: G sets (Session id == Null to 'M') and shares this connection termination identifier declaration to
    other members of the cluster and base station (M-Malicious)
-----------------------------------------------------------------

**Figure 9** **The pseudo-code of Algorithm 5.**

is rather complicated to be detected. However, the nodes' signatures and traffic patterns help identify wormhole attackers in tactical WSN, which are implemented in the proposed QWDGN system. Algorithm 5 (as shown in Fig. 9) describes the procedure of wormhole detection.

At the end of this process, various wormholes can be detected. In this work, a secure Ad Hoc On-Demand Distance Vector (AOMDV) (*Fathi et al., 2021*; *Sartori & Melen, 2021*) was used for a safe routing process, in which the transmission path is declined once the attackers are identified successfully. Then, the path is forever marked as vulnerable in the AOMDV routing table, thus ensuring secure routing.

## IMPLEMENTATION AND RESULTS

Herein, the proposed QWDGN was verified using Network Simulator-3 (NS-3). The network was designed with the simulation characteristics, which are provided are in Table 2. In this simulation, 300 tactical WSN nodes were created with an initial transmitting power of 0.5 Joules, then the tactical WSN area of 1000*1000 m was configured.

Figure 10 shows the performance of the proposed QWDGN system at different security levels of links, where the link counts vary according to different HMAC lengths. For this performance analysis, three existing wormhole detection techniques, namely wormhole detection in static WSN (WSWSN), round-trip time (RTT) based wormhole detection (RTTWD), and AODV for wormhole detection (AWD), were included for comparison. In the WSWSN approach, several wormholes are simulated on a static WSN and are identified using signature-based procedures. However, this approach assumes that all nodes are preconfigured with predefined guard nodes (*Patel, Aggarwal & Chaubey, 2019*).

Figures 11 and 12 present the details of the different wormhole detection methods, namely WSWSN, RTTWD, and AWD, compared with the proposed QWDGN system.

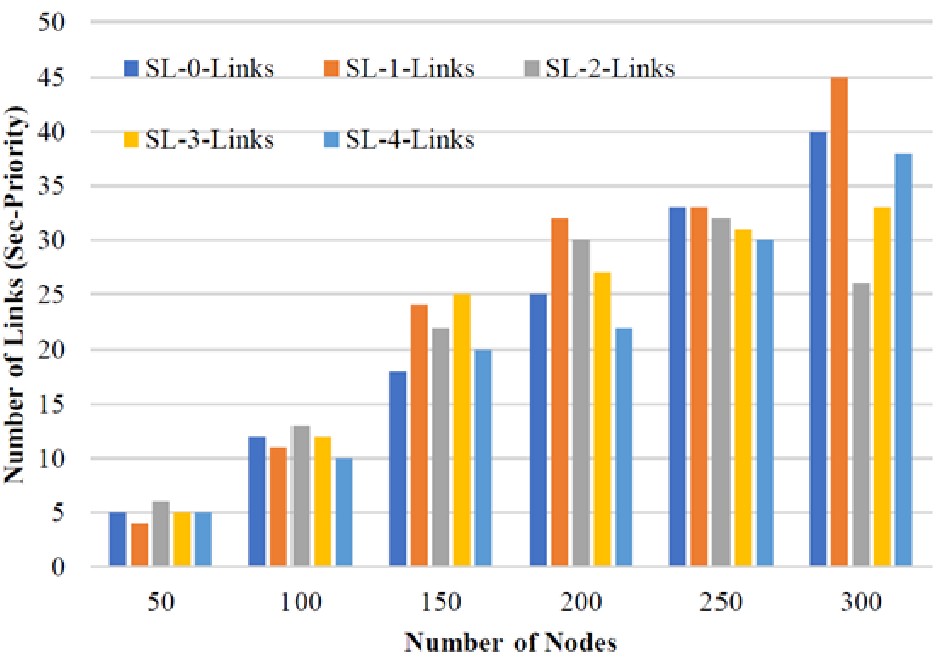

**Figure 10** Nodes and secure links.

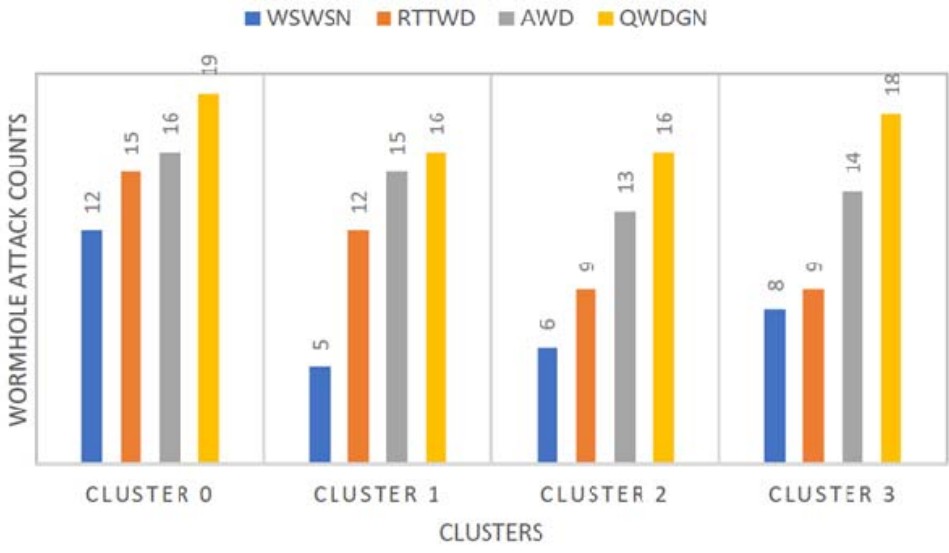

**Figure 11** Number of attacks detected.

Results show that QWDGN performs better than the other methods due to its dynamic QoS aware guard nodes and security aspects.

Figure 13 illustrates the comparison of QoS-based overhead between our proposed strategy and the other methods. In this case, the different methods lack quality parameters

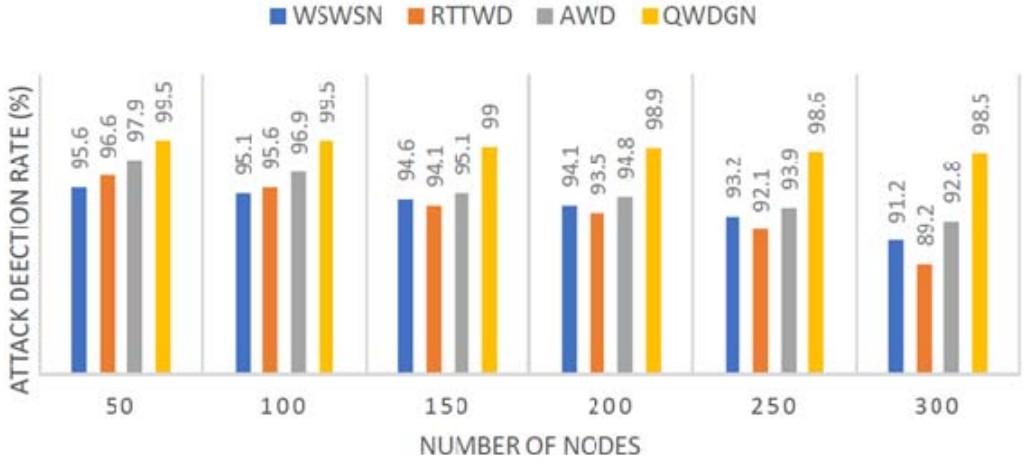

**Figure 12  Attack detection ratio.**

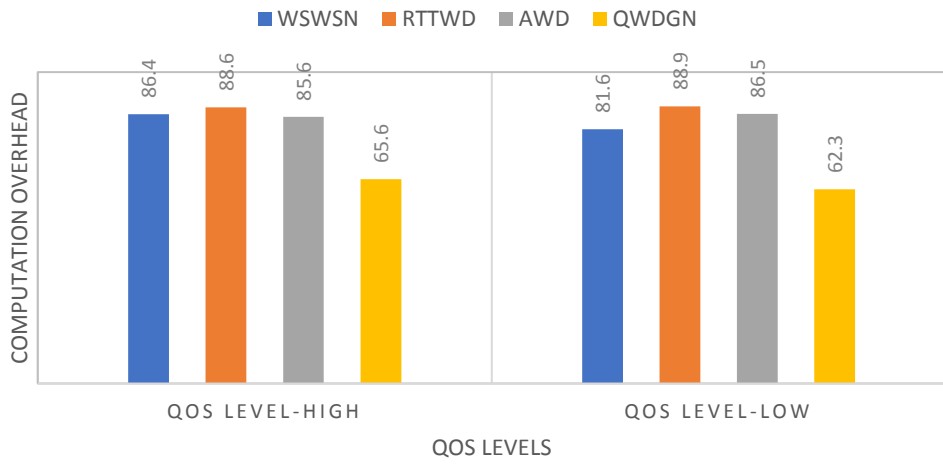

**Figure 13  QoS levels and computation overhead.**

of the sensitive tactical WSN, which increases the complexity of all QoS levels when developing security principles. Comparatively, our proposed QWDGN system ensures limited overhead in providing security features to all links.

## CONCLUSION

The proposed QWDGN was designed and implemented to create a wormhole detecting guard node environment. The nodes were selected based on the required quality levels of each link, which were detained separately as priority based on QoS needs. In this environment, the selected guard nodes were dynamically switched to detect the wormholes considering QoS needs, while routing was carried out using secure AOMDV. According to the performance evaluation, the proposed strategy can produce a sufficient number of

guard nodes and ensure security of the tactical WSN based on the QoS provision levels with 20% less overhead. Nevertheless, the main challenge of this proposed system is maintaining the availability of guard nodes against dynamic network changes. Thus, our strategy can be improved by using secure quality aware routing algorithms to minimize the latency by 20% and packet loss and implementing deep learning-based neural network techniques to increase the dynamic decision-making rate against wormholes.

### Funding
The author received no funding for this work.

### Competing Interests
The author declares that there are no competing interests.

### Author Contributions
- Ke Zhang conceived and designed the experiments, performed the experiments, analyzed the data, performed the computation work, prepared figures and/or tables, authored or reviewed drafts of the article, and approved the final draft.

### Data Availability
The raw data and code are in the Supplemental Files.

### Supplemental Information
Supplemental information for this article can be found online at http://dx.doi.org/10.7717/peerj-cs.1449#supplemental-information.

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
