# Peer review of "Retraction Notice"

_PeerJ Computer Science, doi:10.7717/peerj-cs.1449_

## Round 0.1 · original submission · Major Revisions

According to the reviewer comments and my suggestion, the manuscript needs to be further improved. The reviewers questioned the current introduction part, and the author should further optimize it. In addition, some reviewers pointed out that some concepts and descriptions need to be further clarified. Some review comments are aimed at the resolution of pseudocode, and the author should also pay enough attention. Please carefully address the above concerns.

Reviewer 1 ·

Basic reporting

There are some ambiguous sentences
The result for me is not clear. In my opinion the researcher should clarify the result and how he compare it with other result .
"can produce a
219 sufficient number of guard nodes and ensure security of the tactical WSN based on the QoS provision
220 levels with 20% less overhead. Nevertheless, the main challenge of this proposed system is maintaining "
How the researcher know about the 20%

Why the researcher choose this attributes " 201 Table 2. Simulation attributes"

Experimental design

The aim clear
the methods need more details

Validity of the findings

The experimental need to write in detail
how did he calculate the 20%
show the result in details

Cite this review as

Reviewer 2 ·

Basic reporting

The proposed measures of the QoS-based security provision, including link cluster formation, guard node selection, authenticated guard node identification, and intrusion detection, ensure economic and efficient network communication with different quality levels. The novelty and contribution is good. However, there are some concerns need to be addressed.

Experimental design

The security of wireless sensor networks has always been a research topic that has attracted much attention in academia. A secure wireless sensor network is very important for applications. This paper proposes a QAS-based link management strategy to resist wormhole attacks. A large number of experimental results show that the method proposed in this paper has certain effectiveness. The writing and innovation of this article are acceptable, but there are still some problems that need to be resolved.

Validity of the findings

1. In Introduction, in addition to introducing the work of predecessors, the author should also summarize what is the work of this paper? For example, how does completeness combine with QoS? In addition, the thesis structure of this paper also needs to be further described clearly.
2. In section II, is the security of the nominal computational overhead guaranteed?
3. I think the authors should further explain Figure 1.
4. In addition to the optimal protection node selection algorithm, does this paper also study secure cluster head identification and trust evaluation procedures. If so, the author needs to clarify further.
5. The authors need to supplement the comparison between the algorithm in this paper and the reference algorithm in terms of QoS-based calculation load.

Additional comments

See above

Cite this review as

·

Basic reporting

Wireless Sensor Networks (WSN) is a distributed sensor network whose ends are sensors that can perceive and inspect the outside world. The sensors in the WSN communicate wirelessly, so the network setting is flexible, the location of the device can be changed at any time, and it can also be connected to the Internet in a wired or wireless manner. A multi-hop self-organizing network formed by wireless communication. This paper attempts to study link management in tactical wireless sensor networks. The innovation and writing of this article are good, but there are many problems that need to be revised.

I think the experiments in this article are relatively rich, but the author lacks corresponding explanations for some results. I think the author should make it clearer in the revised manuscript.
1. Although wireless sensor networks have been studied for many years, I think the author should clarify its definition in the abstract.
2. This paper studies link management, but I don't seem to find the author's description of the dynamics of wireless sensor networks in the article. For example, whether changes in environmental conditions cause changes in the bandwidth of the wireless communication link.
3. I think the author lacks a corresponding description of the amount of information that Figure 1 wants to express.
4. The pseudo-code of the algorithm in this article is too vague, and I think the resolution can be further improved.

Experimental design

.

Validity of the findings

.

---

## Round 0.2 · accepted · Accept

After my reading of the revised manuscript, I can recommend it for acceptance. All of concerns raised by reviewers have been addressed well.